# Comparison of Toxicity and Cellular Uptake of CdSe/ZnS and Carbon Quantum Dots for Molecular Tracking Using *Saccharomyces cerevisiae* as a Fungal Model

**DOI:** 10.3390/nano14010010

**Published:** 2023-12-19

**Authors:** Sanni M. A. Färkkilä, Monika Mortimer, Raivo Jaaniso, Anne Kahru, Valter Kiisk, Arvo Kikas, Jekaterina Kozlova, Imbi Kurvet, Uno Mäeorg, Maarja Otsus, Kaja Kasemets

**Affiliations:** 1Institute of Ecology and Earth Sciences, University of Tartu, Juhan Liivi 2, 50409 Tartu, Estonia; 2Laboratory of Environmental Toxicology, National Institute of Chemical Physics and Biophysics, Akadeemia tee 23, 12618 Tallinn, Estonia; monika.mortimer@kbfi.ee (M.M.); anne.kahru@kbfi.ee (A.K.); imbi.kurvet@kbfi.ee (I.K.); maarja.otsus@kbfi.ee (M.O.); 3Institute of Physics, University of Tartu, W. Ostwaldi 1, 50411 Tartu, Estonia; raivo.jaaniso@ut.ee (R.J.); valter.kiisk@ut.ee (V.K.); arvo.kikas@ut.ee (A.K.); jekaterina.kozlova@ut.ee (J.K.); 4Institute of Chemistry, University of Tartu, Ravila 14a, 50411 Tartu, Estonia; uno.maeorg@ut.ee

**Keywords:** quantum dots, carbon dots, mycorrhizal fungi, nutrients, molecular tracking, *Saccharomyces cerevisiae*, nanoparticle uptake, nanoparticle toxicity, viability

## Abstract

Plant resource sharing mediated by mycorrhizal fungi has been a subject of recent debate, largely owing to the limitations of previously used isotopic tracking methods. Although CdSe/ZnS quantum dots (QDs) have been successfully used for in situ tracking of essential nutrients in plant-fungal systems, the Cd-containing QDs, due to the intrinsic toxic nature of Cd, are not a viable system for larger-scale in situ studies. We synthesized amino acid-based carbon quantum dots (CQDs; average hydrodynamic size 6 ± 3 nm, zeta potential −19 ± 12 mV) and compared their toxicity and uptake with commercial CdSe/ZnS QDs that we conjugated with the amino acid cysteine (Cys) (average hydrodynamic size 308 ± 150 nm, zeta potential −65 ± 4 mV) using yeast *Saccharomyces cerevisiae* as a proxy for mycorrhizal fungi. We showed that the CQDs readily entered yeast cells and were non-toxic up to 100 mg/L. While the Cys-conjugated CdSe/ZnS QDs were also not toxic to yeast cells up to 100 mg/L, they were not taken up into the cells but remained on the cell surfaces. These findings suggest that CQDs may be a suitable tool for molecular tracking in fungi (incl. mychorrhizal fungi) due to their ability to enter fungal cells.

## 1. Introduction

Mycorrhizae, the ancient symbiosis between plant roots and fungi, have risen to the knowledge of the general public as altruistic conduits through which nutrients are moved around between forest trees and weak or young trees are nurtured by the old and strong ones [1,2]. While nutrient transfer in mycorrhizal systems has been studied isotopically for a relatively long time [3,4], and some studies have indicated the potential movement of nutrients between plants connected via the mycorrhizae, several scientists have recently raised concerns about this narrative [1,2,5]. Indeed, based on a recent meta-analysis [5], previous results show little evidence of this kind of nutrient transfer. One major reason is that testing the nutrient transfer with isotopic methods does not enable the exclusion of alternative pathways, as the transfer pathway is only deduced based on various control treatments instead of being directly observed [2,5]. In this respect, conjugating the nutrients to intrinsically fluorescent quantum dots (QDs) for nutrient tracking in mycorrhiza-plant systems could be a viable option to exclude the alternative pathways and determine whether such transfer is taking place [5]. Unlike isotopically labelled nutrients, QDs conjugated with nutrients (such as an amino acid or apatite) can be visually followed to determine transfer pathways in real-time instead of just measuring total isotope concentrations in the final target after exposure [6,7]. It has even been reported that CdSe/ZnS semiconductor QDs conjugated to a nitrogen source could be tracked from the soil through mycorrhizal fungi to plants, even in field conditions [8,9]. Thus, QDs are a potential alternative technique for studying plant-to-plant nutrient transfer via mycorrhizae. 

However, their potential toxicity hinders the application of cadmium-containing QDs in ecological studies [10,11,12,13,14,15,16]. Many studies have reported toxic effects of such QDs, as Cd is leached from the particle core, inducing adverse responses [12,14,16]. Indeed, many authors have expressed the need for developing more biocompatible QD alternatives that are either stabilized by a surface coating (e.g., ZnS) that hinders Cd release from the core of QDs or are prepared from other components than Cd [17,18,19,20,21,22]. To address the issue, developing new types of QDs and improving their properties is ongoing. While most biological studies still utilise semiconductor QDs that contain Cd, more biocompatible options, such as carbon quantum dots (CQDs), are becoming more common [23]. The availability of such new and improved QD varieties gives hope for solving biological riddles that have remained unsolved because Cd-containing markers are not applicable due to ecosafety reasons. In this respect, an obvious advantage of CQDs compared to Cd-containing QDs is that they comprise carbon, oxygen, and nitrogen, all elements regularly present in living organisms and biological molecules. 

The goal of this study was to find an alternative type of QDs (free of toxic elements) that could be used for tracking nutrients in plant-fungal systems analogously to previously used CdSe-based QDs. For that, we synthesised amino acid-based CQDs and compared them to previously used amino acid-conjugated CdSe/ZnS QDs, using the single-cell budding yeast *Saccharomyces cerevisiae* as a model organism. Amino acids represent a nitrogen source relevant in the nutrient transfer between forest trees, as nitrogen is usually the limiting nutrient in those environments and is considered the main benefit of mycorrhizae in such conditions [24,25]. In this context, the amino acid cysteine (Cys) was selected because cysteine-conjugated CdSe/ZnS QDs were previously reported to be taken up in relatively large quantities by mycorrhizal fungi [26]. *S. cerevisiae* is an advantageous model organism due to its short life cycle and ability to grow in controlled conditions [27]. Furthermore, as it is well studied, a lot of information related to its lifestyle, molecular uptake, and suitable bio-imaging techniques is already available [28,29]. Tracking nutrient transfer in fungi with QDs requires that QDs with their ‘cargo’ will be internalised into fungal cells. Like mycorrhizal fungi, *S. cerevisiae* possesses a rigid cell wall that may restrict the entry of particles into its cells. It is generally accepted that mycorrhizal fungi use endocytosis to take up large particles, analogously to yeast cells that use clathrin-mediated endocytosis [30], and Au, polystyrene, and Ag nanoparticle uptake by yeast cells via endocytosis has been shown [31,32,33]. Therefore, QDs uptake by *S. cerevisiae* can serve as a proxy for more complex fungi, such as mycorrhizal fungi.

A review of the literature on *S. cerevisiae* and other commonly studied yeasts and their interactions with QDs (Appendix A) showed that 18 studies report toxic effects of Cd-containing QDs on yeast, while only one concludes that there are no harmful effects. Conversely, reports of CQDs toxicity are split, with three reports of no toxicity and three of toxic effects. In terms of uptake, eight Cd-QDs studies report that the particles enter cells, while another eight report no entry into cells. On the other hand, six out of nine CQDs papers report particles entering cells, while two report no uptake, and one notes that uptake depends on particle precursors. These data indicate that the effects and uptake of QDs are far from clear, even in the case of a relatively simple and extensively studied fungal model such as yeast. Thus, more experiments at this level of complexity are needed before one can move on to experiment with the much more complicated mycorrhizal systems in which several species of divergent taxonomies interact.

## 2. Materials and Methods

### 2.1. Quantum Dots (QDs), Conjugation of Cysteine to QDs, and Physicochemical Characterization

Commercial carboxyl-capped CdSe/ZnS quantum dots (QDot ITK 585, Invitrogen, Carlsbad, CA, USA) were tested as purchased, and conjugation to L-cysteine (97%, Sigma-Aldrich, St. Louis, MO, USA) was performed according to the manufacturer’s protocol. A 10 mg/mL solution of cysteine diluted in 10 mM borate buffer (pH 7.4) was used for the reaction.

The CQDs were prepared according to the slightly modified protocol of Suner et al. [34]. A portion of 0.57 g (4.7 mmol) of L-cysteine was dissolved in 10 mL of deionised water (Milli-Q, Merck Millipore, Burlington, MA, USA). After adding 1.5 g (7.8 mmol) of citric acid (Lachner, analytical grade), the mixture was ultrasonicated for 10 min. The solution was treated in a Philips microwave oven at 900 W for 4 min. The remaining material was washed with 10 mL of deionized water at room temperature and filtered through a 0.22 µm pore-size membrane. Then, the filtrate was evaporated under vacuum and at room temperature to dryness. The solid material prepared was dissolved in MilliQ water, freeze-dried, and manually ground into a powder. 

To measure the optical spectra of the QDs, the aqueous solutions were prepared in a 10 mm pathlength quartz cuvette. The absorbance spectra were recorded using a Jasco V-570 spectrophotometer (Easton, MD, USA), and the fluorescence (excitation and emission) spectra were acquired by a Jobin-Yvon Horiba Fluoromax 4 spectrofluorometer (Irvine, CA, USA). The latter used a 150 W xenon arc lamp for excitation and a photomultiplier (photon counting) for detection. The spectra were corrected against the spectral response of detection, the spectral distribution of excitation, and the finite absorbance of the sample. The spectral bandwidths of the excitation and emission monochromators were 2 nm or less.

Fourier transform infrared (FTIR) spectroscopy of QDs was conducted using a Spectrum BXII FTIR spectrometer (Perkin Elmer, Waltham, MA, USA) with the attenuated total reflection (ATR) technique with a ZnSe crystal. The spectral range was 4000–600 cm^−1^ and the resolution was 4 cm^−1^.

Transmission electron microscopy (TEM) and scanning TEM (STEM) analysis were conducted on the FEI Titan Themis 200 system (Thermo Fisher Scientific, Waltham, MA, USA) operated at 200 kV. Both bright (BF) and high-angle annular dark field (HAADF) images were recorded in STEM mode. X-ray photoelectron spectroscopy (XPS) measurements were conducted at ultra-high vacuum conditions using an electron energy analyser SCIENTA SES-100 (Uppsala, Sweden) and Mg K_α_ X-rays from a non-monochromatic twin-anode X-ray tube (Thermo XR3E2, Waltam, MA, USA). For the preparation of QD samples for TEM and XPS, the suspension of the QDs in isopropanol was drop-cast onto a thin carbon film or silicon substrate, respectively, and dried. 

The hydrodynamic size and zeta potential (a proxy for surface charge) of the QDs were measured using a ZetaSizer Nano ZS instrument (Malvern Instruments Ltd., Worcestershire, UK). For the measurement, QDs stock suspensions (1500 mg/L, 12,000 mg/L, and 4200 mg/L; for CQDs, pristine CdSe/ZnS QDs and CdSe/ZnS conjugated to cysteine, respectively, all in 50 mM borate buffer, pH 8.3) were probe sonicated for 5 min, then diluted in MilliQ water to the range of 100–500 mg/L and measured immediately.

### 2.2. Saccharomyces Cerevisiae BY4741 and Cultivation Conditions

Laboratory strain *S. cerevisiae* BY4741 (*MAT*α; *his*3*Δ*1; *leu*2*Δ*0; *met*15*Δ*0, *ura*3*Δ*0) (EUROSCARF at the Institute of Microbiology, University of Frankfurt, Frankfurt, Germany) was used in this study. Frozen permanent copies were stored at −80 °C. The master agar plate containing yeast-peptone-dextrose (YPD) medium was inoculated with the frozen culture and incubated for 72 h at 30 °C. The YPD comprised 1% yeast extract (Lab M, Bury, UK), 2% Bacto-peptone (Gibco, Billings, MT, USA), 2% glucose (Sigma-Aldrich, St. Louis, MO, USA), and 1.5% agar (Neogen, Lansing, MI, USA) for agar medium. An overnight culture was initiated from the 2–3 colonies on the master plates and incubated in YPD at 30 °C for 18–22 h at 200 rpm (revolutions per minute). The mid-exponential phase culture was started from the overnight culture with an initial optical density at 600 nm (OD_600_) of 0.25 (measured using a Yenway 6300 Spectrophotometer (UK) with 1 cm polystyrene cuvettes) and cultivated in YPD at 30 °C and 200 rpm until the OD_600_ of ~1 (approximately 4 h).

### 2.3. Toxicity Testing of QDs (Spot Test)

The toxicity of the QDs against the yeast *S. cerevisiae* BY4741 was assessed in MilliQ water, following the protocol described in Suppi et al. [35]. In brief, yeast cells from the mid-exponential phase culture, grown in YPD liquid medium at 30 °C, were harvested by centrifugation at 4126× *g* for 5 min (Sigma 3-16KL, Sigma, Osterode am Harz, Germany), washed twice with MilliQ water through centrifugation, and then resuspended in MilliQ water to an OD_600_ of 1.2 (corresponding to ~2 × 10^7^ colony-forming units or CFU/mL). Subsequently, 75 µL of this yeast culture was added to the wells of a 96-well microplate (BD Falcon 351172, Fisher Scientific, Hampton, NH, USA); each well contained either 75 µL of MilliQ (as a negative control) or QDs dilutions in MilliQ water. QDs concentrations of 0.1, 1, 10, and 100 mg/L were tested. Additionally, AgNO_3_ was studied as a positive control. The microplates were then incubated for 24 h at 30 °C without shaking. After 24 h incubation, 5 µL from each well were pipetted/spotted onto YPD agar plates, and the plates were incubated in the dark for 72 h at 30 °C. Finally, colony formation was visually assessed to determine each tested compound’s minimum biocidal concentration (MBC), i.e., the lowest concentration inhibiting colony formation on the toxicant-free nutrient-agar plates. The experiments were conducted twice, each with four replicates per test.

### 2.4. Evaluation of the QDs Cellular Interactions by Confocal Laser Scanning Microscopy

For the assessment of QDs interactions with yeast cells by confocal laser scanning microscope (CLSM), yeast cells were grown and harvested as described in Section 2.3 and then stained with 2× CellBrite Fix 555 membrane stain (Biotium, Fremont, CA, USA) for 15 min in the dark. Cells were centrifuged at 9300× *g* for 1 min (Eppendorf Centrifuge 5415 R, Hamburg, Germany) to remove the stain and then resuspended in MilliQ water (final OD_600_ ~1). Next, 500 µL of stained cells were exposed to CdSe/ZnS QDs (1 mg/L), CdSe/ZnS-Cys QDs (1 mg/L), and CQDs (100 mg/L) at room temperature for 24 h in 1.5 mL microtubes (Nerbe Plus, Winsen, Germany).

After incubation, to remove the loosely bound QDs, half of the cell suspension was transferred to a new 1.5 mL microtube, centrifuged for 1 min at 9300× *g*, the supernatant removed, and yeast cells resuspended in MilliQ water. Then, 20 µL of the cell suspension from each tube, i.e., washed and unwashed yeast cells, was pipetted on a microscopy slide (Deltalab, Barcelona, Spain) and dried at 37 °C for 20 min. Samples were mounted in 10 µL of mounting medium Mowiol (Mowiol^®^ 4-88, Sigma Aldrich, Darmstadt, Germany) and covered with an 18 × 18 mm coverslip (Kaltek, Villatora, Italy). The cellular interactions of QDs were visualised using the Zeiss LSM800 CLSM (Germany) with a 100x oil immersion objective. The excitation/emission track settings were as follows: for the CellBrite 555 signal, 561/530–700 nm; CdSe/ZnS QDs and CdSe/ZnS-Cys 405/540–620 nm; and CQDs, 405/400–530 nm. Images were processed using ZEN software (Carl Zeiss Microscopy, Oberkochen, Germany).

## 3. Results

### 3.1. Physico-Chemical Characteristics of Quantum Dots

The absorbance and fluorescence spectra of CdSe-based QDs corresponded to the characteristics given by the manufacturer (wide absorbance and relatively narrow fluorescence band at 585 nm; see Appendix A). For CQDs, the optical measurements showed that the absorption band has a maximum of 350 nm and a fluorescence emission of around 430 nm (Figure 1). The absorbance at 350 nm has been assigned to the n-p* transition of the surface state [36]. The optical spectral properties of CQDs strongly depend on the preparation route and precursors, (see, e.g. [37]), whereas frequently another type of absorbance with shoulders in the UV region (<350 nm) is observed, (see, e.g. [38]).

The fluorescence excitation spectra correspond well to the absorption. The fluorescence emission peak of our CQDs did not shift noticeably when the excitation wavelength changed, which indicates the uniformity of the synthesized carbon dots both in terms of size and composition. The spectra were almost identical when measured approximately six months apart, indicating that CQDs are stable and retain their optical properties even after long periods. The quantum yield of CQDs fluorescence was determined to be 65 ± 5% by comparison with the quinine sulfate reference [39], prepared as 10^−5^ M of dye in a 0.5 M H_2_SO_4_ solution. 

The FTIR spectrum of prepared CQDs is shown in Figure 2. The very broad signal in the region of 3650–2200 cm^−1^ demonstrated the stretching vibration of the -OH, and -NH groups. This also covers the N-H bending and C=O stretching Fermi resonance signals. The band at 2590 cm^−1^ is characteristic of S-H stretching vibration, and the peak at 1698 cm^−1^ corresponds to the C=O bond stretching. The shoulder of the C=O signal at 1650 cm^−1^ could belong to NH bending (amid II) vibration. The peaks at 1519, 772, and 685 cm^−1^ indicate the presence of an aromatic system, and the 1393 cm^−1^ peak is characteristic of C-N vibration. Signals at 1204 and 1175 cm^−1^ are related to C-S and C-O stretching vibrations, respectively [40] These observations indicate that the CQDs synthesis reaction between citric acid and cysteine was successful.

TEM and STEM analysis of the CQDs samples shows agglomerates with sizes around 20–40 nm (Figure 3). Occasionally, larger agglomerates with sizes up to 1 mm can be observed. The high-resolution TEM and STEM HAADF images clearly show the presence of a substructure with a feature size of 2–3 nm (Figure 3B,D) inside the larger particles. These more minor features have a characteristic size, which can be associated with blue fluorescence emission [41]. No crystal fringes can be observed, indicating that the structure is amorphous.

The TEM images of CdSe/ZnS-Cys QDs show relatively large aggregates consisting of crystalline regions with a typical size of 5 nm (Figure 4). The Fourier transform image of the crystalline region characterizing the reciprocal lattice is given in the Appendix A. The surrounding shell of the QDs in Figure 4B is blurred and not clearly visible on the carbon film of a TEM grid. 

The XPS measurements were conducted to identify the elemental composition of QDs. As expected, the survey spectrum (Figure 5A) shows the presence of Cd and Zn in the pristine CdSe/ZnS QDs. However, in the spectra of CdSe/ZnS-Cys QDs, the lines of metal elements are entirely absent, indicating that the layer added during the conjugation procedure is thick enough to obscure the signal from the semiconductor core/shell materials. In all spectra, C and O lines are present, which can be related to the organic layer on CdSe/ZnS QDs and is intrinsic to CQDs. 

To assess the presence of nitrogen on the QDs surface, we took a closer look at the N1s peaks (Figure 5B). The N1s peaks appear at different energies, which indicates different types of nitrogen bonding in carbon and semiconductor QDs. The CdSe/ZnS-Cys N1s energy is closest to that of the bond in cysteine [42].

The manufacturer-reported diameter of the commercially obtained pristine CdSe/ZnS QDs was ~10 nm, which would increase to ~15–20 nm after conjugating biomolecules to the surface of the QDs. Expectedly, the hydrodynamic diameters of both types of QDs were larger than the pristine sizes due to the agglomeration of particles and the formation of the hydration layer on the particle surface in aqueous media. The size distributions of CdSe/ZnS and CdSe/ZnS-Cys QDs based on the scattering intensity showed the presence of larger particles/agglomerates. However, volume-based distribution indicated that the smaller-sized fraction (20.8 ± 7.1 and 14.3 ± 3.6 nm, respectively) dominated in the suspensions (Figure 6A,B). On the other hand, CQDs had a relatively small average hydrodynamic diameter of ~6.08 nm (Table 1). Despite a quite wide size distribution based on the scattered light intensity, smaller particles comprised most of the suspension composition (Figure 6C). These results indicate that while all particles had a similar proneness to aggregation (PDI ~0.4–0.5), the average hydrodynamic size of CQDs was significantly smaller than that of the commercial CdSe-based QDs. All particles were negatively charged in MilliQ water, as indicated by zeta potential (Table 1). While the CdSe/ZnS QDs were highly negatively charged, with a slightly reduced absolute value of zeta potential after cysteine conjugation, the CQDs had a significantly smaller absolute zeta potential. Still, the zeta potential of CQDs was close to −20 mV, which is considered to provide sufficient electrostatic repulsion between the particles, thus ensuring a relatively stable particle suspension. 

### 3.2. QDs Were Not Toxic to Yeast S. cerevisiae

The viability assay (spot test) revealed that each of the three varieties of QDs tested (CdSe/ZnS, CdSe/ZnS-Cys, and CQDs) had an MBC >100 mg/L after 24 h of exposure, as even after exposure to QDs at a 100 mg/L concentration, yeasts were able to yield typical colonies on toxicant-free nutrient agar (Figure 7). Conversely, the MBC for AgNO_3_ (a well-known antimicrobial agent tested as a positive control) was as low as 0.5 mg Ag/L. These results indicate that none of the QDs were toxic to *S. cerevisiae*, at least in concentrations necessary for assessing their uptake and visualisation using the CLSM.

### 3.3. CQDs Readily Entered Yeast Cells, While CdSe/ZnS QDs Did Not

The uptake of QDs by *S. cerevisiae* was assessed under a CLSM. Concentrations of 1 mg/L for CdSe/ZnS QDs (with and without cysteine) and 100 mg/L for CQDs were experimentally selected for microscopic visualisation. The examination revealed that semiconductor QDs, regardless of whether they were conjugated to cysteine or not, were observed on the surfaces of yeast cells but not inside them (Figure 8B,C). Washing the cells after exposure reduced the visible signal from the particles to close to none (Figure 8F,G). This result indicates that these particles were only weakly bound to the surfaces of the yeast cells and did not enter them even after a relatively long exposure time (24 h). CQDs were observed to be taken up by yeast cells, and particles appeared to distribute throughout the cytoplasm (Figure 8D). The signal inside the cells was preserved after washing (Figure 8H). 

## 4. Discussion

In this study, CQDs were successfully synthesized, as indicated by the analysis of particle properties. The optical measurements revealed that as-prepared CQDs have effective, stable fluorescence even after long periods. FTIR analysis also confirmed that the synthesis reaction was successful, as nitrogen and sulphur from the cysteine were observed in the final product. The synthesised CQDs have a narrow fluorescence peak, as is typical for QDs. However, contrary to the tested CdSe/ZnS QDs, the CQDs emission peak is in the blue range. Excitation with ultraviolet (UV) light, which is needed for the efficient emission of QDs, typically also causes blue autofluorescence of plant cell wall components such as lignin, suberin, and cutin, which can make separating the QDs signal within plants and mycorrhizae challenging [43]. Yet, as the quantum yield is relatively high (65 ± 5%) and the CQDs fluorescence lifetime exceeds those typical of autofluorescence (data not shown), this issue can be bypassed by comparing to non-labelled controls and/or time-gating (capturing the QDs fluorescence signal after a short-lived autofluorescence signal has disappeared) [23]. The fluorescence emission of CdSe/ZnS QDs is in the orange range, as reported by the manufacturer. Thus, the signal from these QDs is usually easier to separate from plant cell wall autofluorescence. However, this comes at the cost of particle size since, owing to quantum confinement effects, the size of CdSe/ZnS QDs increases with increasing emission wavelength [18]. Indeed, the dynamic light scattering (DLS, performed with the Malvern Zetasizer) and TEM analysis point to CQDs having a main particle fraction with a smaller size than CdSe/ZnS QDs. As nanoparticle uptake is strongly linked to particle size [44], this trade-off in favour of emission in a longer wavelength range may not be worth it in fungal molecular tracking applications. 

The toxicity assay showed that none of the tested QDs were toxic at the tested concentrations (up to 100 mg/L). This indicates that, in terms of toxicity, neither of the particles was better or worse for potential molecular tracking applications in yeast. The Cd-containing QDs did not exert toxicity on *S. cerevisiae*, probably due to the presence of the ZnS shell, which hindered Cd release from the particles. Several studies have shown that this significantly reduces particle toxicity compared to similar particles that do not have a shell [21]. A similar trend was evident in the results summarised in Appendix A, as the reported toxicity was generally lower in the case of Cd-containing particles with the ZnS shell than particles without it. Nonetheless, grounds for avoiding Cd-containing particles still exist, as they may become more problematic long-term when the particle starts to degrade [14,15,16]. Furthermore, as with the achievement of longer emission wavelengths, adding protective shells is also a trade-off with size, since this inevitably increases the particle diameter. CQDs, on the other hand, are generally considered safe as they contain no hazardous elements [45,46,47]. There are even reports of the positive effects of low concentrations of CQDs [48,49,50,51]. Comparative toxicity studies between Cd-containing QDs and CQDs are still relatively rare and often hindered by a lack of information on particle masses, which are needed so that the toxicity assessment can be made in comparable units [23]. Still, comparative studies in animal cells and microalgae have shown CQDs to be significantly more biocompatible than Cd-containing QDs [52,53]. Indeed, while some reports of CQDs toxicity exist, the concentrations noted to induce adverse effects are usually over 100 times larger than those reported for Cd-containing QDs (Appendix A). However, the toxicity of CQDs, like Cd-containing QDs, depends on the particle properties, precursors, and modifications [54,55]. Thus, toxicity testing of a particular type of CQDs should be performed on the desired target, as the diversity of synthesis pathways and resulting particles categorized as CQDs is vast [55,56,57].

CLSM is an advantageous method for observing particle uptake; its relatively simple and cost-effective sample preparation allows for microscopic image stacking that enables the differentiation between nanoparticles adsorbed to cell surfaces and nanoparticles in cells [29]. Based on the image stacking, we confirmed the entry of CQDs into yeast cells. This observation aligns with several previous reports regarding CQDs (Appendix A) and indicates that CQDs are a promising tool for nutrient tracking in fungi. Endocytosis, or passive transport across cell membranes [44,58], may be responsible for CQD entry. As the CQDs were synthesised with cysteine as a precursor, producing the same exact particle without cysteine is impossible. Thus, it was impossible to test whether the presence of the nitrogen source played a role in the uptake of the particle or not. Based on the reviewed literature, CQDs from various precursors may enter the cells, which makes unspecific uptake likely. More research is needed to determine the level of specificity of CQD uptake. However, in terms of exploring transfer within a mycorrhizal network, observing the transfer pathway of the particles, regardless of whether they are moving passively or acting as a carbon or nitrogen source, would already give more information about the system’s function. 

Our results suggest that, cysteine-conjugated or not, the commercial CdSe/ZnS QDs that were tested did not enter yeast cells. As shown in Appendix A, there are several other reports of various CdSe/ZnS QDs not entering yeast cells. There is also a previous report about cell washing decreasing the fluorescence signal to less than 7% of the signal observed before washing [59]. Thus, our results are in line with previous studies. Still, it is a somewhat unexpected result, considering that these particles have been commercially produced for biotracking purposes, and many studies do report QDs entering cells [60,61,62,63,64]. Plant studies have shown that particles with a positive charge are more readily taken up and translocated than those with a negative charge [65,66,67]. Thus, the strongly negative zeta potential of the commercial QDs, probably caused by the carboxyl capping, which creates COOH groups on the surface of the particles, could impede their uptake. In contrast to the CdSe/ZnS QDs, CQDs had a less negative zeta potential. Additionally, most studies with QDs are performed with animal cells [23,68,69,70], which lack cell walls that may resist the uptake of particles. Indeed, several studies on yeast are utilizing a chemical that makes the cell more permeable [71,72,73] or even experimenting with yeast spheroplasts, from which the cell walls have been removed [74]. Therefore, it is likely that, due to their large size, the Cd-containing QDs that we tested were not able to bypass the cell wall. Some might argue that a possible reason for the lack of uptake of cysteine-conjugated QDs might be that the cysteine is not perceived as useful by the yeast cells and is, therefore, not prompting for the internalisation of QDs. For example, glutathione (GSH)-conjugated QDs have reportedly entered yeast cells [61,62]. Yet cysteine is also a component of GSH, along with two other amino acids. Hence, the two compounds are not very different chemically, and the smaller size of cysteine should make its uptake more efficient. Furthermore, specific transporters for cysteine have been shown to exist in *S. cerevisiae* [75], and its ability to utilise cysteine as a nitrogen source has likewise been previously shown [76]. Thus, there is no theoretical reason to suspect that cysteine-conjugated QDs would be less likely to be taken up via active transport than GSH-conjugated QDs. Therefore, it is more likely that the observed uptake patterns are not related to the composition of the particles or to their conjugates but to the particle diameters. As highlighted before [77], particle size is an important determinant of their uptake and toxicity. Thus, the large aggregate size of the tested CdSe/ZnS QDs likely prevents their internalization, while the smaller CQDs may permeate the cells. This suggests that the large size of the tested CdSe/ZnS QDs may make them unsuitable for molecular tracking in fungi.

## 5. Conclusions

Here we compared self-synthesised CQDs to commercially sourced CdSe/ZnS QDs, with the aim of assessing their suitability for molecular tracking in fungi. Our results revealed that the tested commercial QDs form large aggregates, which have a strongly negative zeta potential. CQDs consisted of several size fractions, but the main fraction had smaller diameters compared to the Cd-containing QDs. CQDs show potential for fungal nutrient tracking, as their small size enables them to enter yeast cells without indication of toxicity. Similarly, the Cd-containing QDs were not toxic. However, the tested CdSe/ZnS QDs did not enter the cells despite being conjugated to a nutrient source. The large aggregate size of these QDs likely prevented them from entering the cells. These results indicate that CQDs may be more suitable for molecular tracking in fungi due to their ability to enter yeast cells. However, more research is needed to determine the specificity of CQD uptake. These results, along with the reviewed literature, serve as a reminder that we still know very little about the interactions of nanoparticles and fungi and that there is a great deal of research needed before these methods can be utilised in complex, natural systems. 

## Figures and Tables

**Figure 1 nanomaterials-14-00010-f001:**
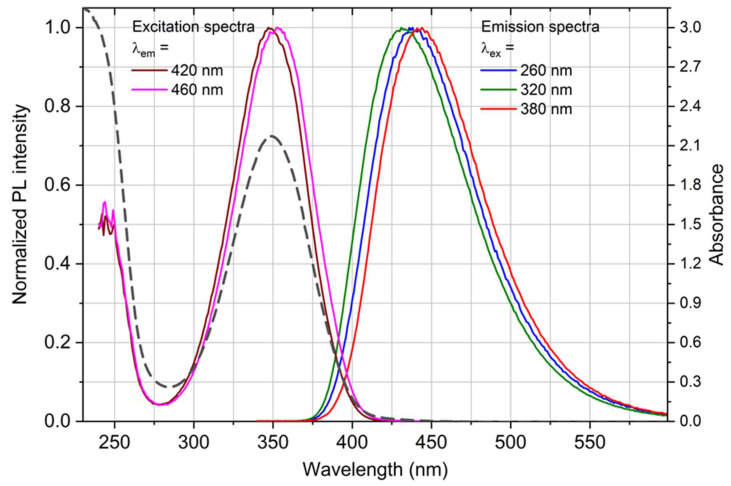
Absorbance spectrum (black dashed line) and normalized fluorescence spectra (solid colored lines) of a 0.29 g/L aqueous solution of carbon quantum dots (PL stands for photoluminescence).

**Figure 2 nanomaterials-14-00010-f002:**
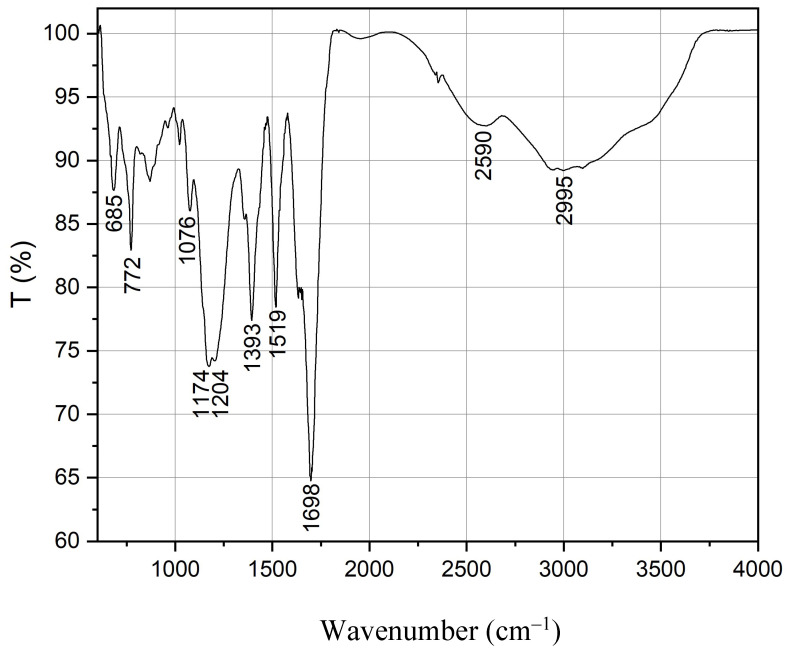
FTIR spectrum of the carbon quantum dots (CQDs) made from L-cysteine and citric acid by microwave technology.

**Figure 3 nanomaterials-14-00010-f003:**
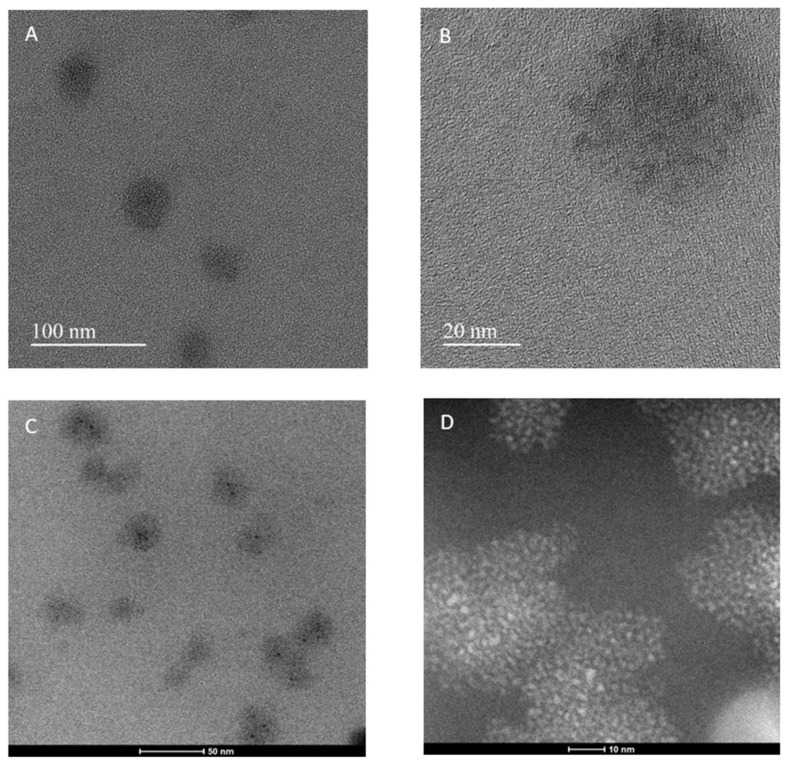
Electronmicroscopic characterization of carbon quantum dots (CQDs). High-resolution TEM (**A**,**B**), STEM BF (**C**), and HAADF (**D**) images of CQDs.

**Figure 4 nanomaterials-14-00010-f004:**
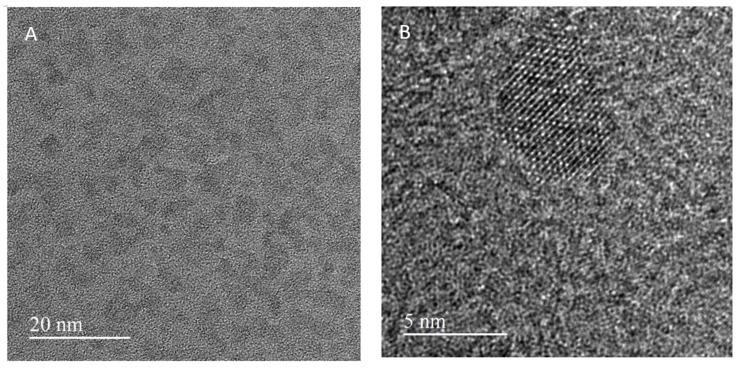
High-resolution TEM images of CdSe/ZnS-Cys QDs with different magnifications (**A**,**B**).

**Figure 5 nanomaterials-14-00010-f005:**
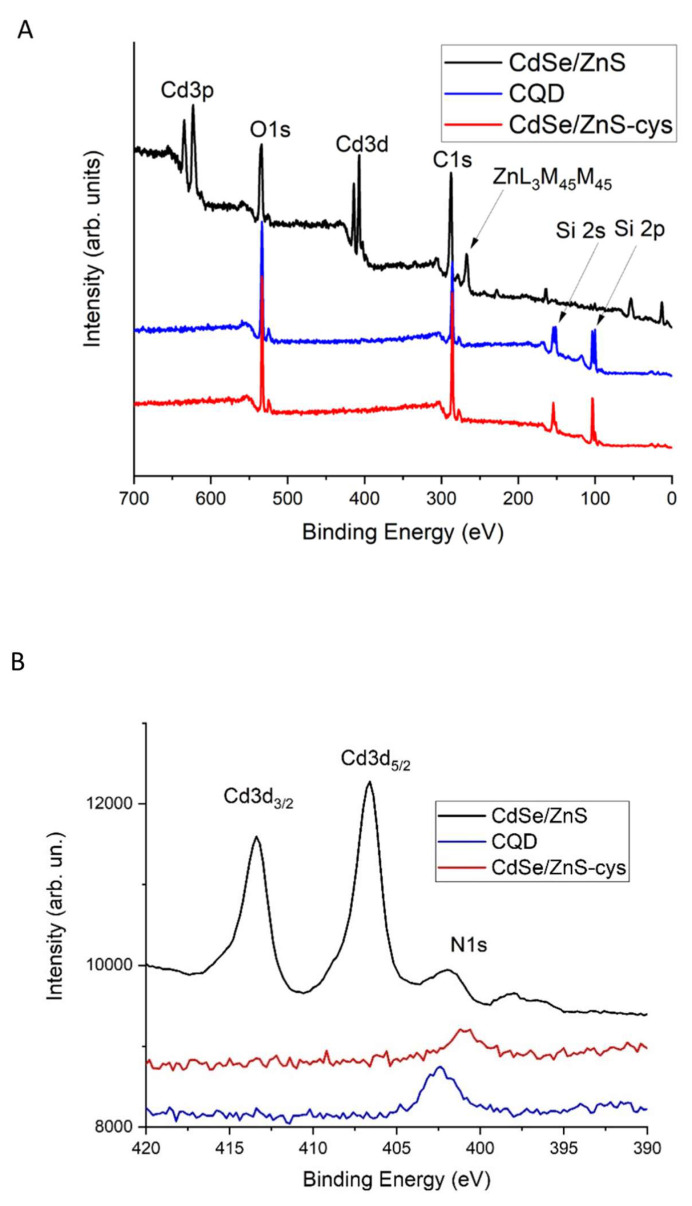
(**A**) The survey XPS spectra of quantum dots. The origins of the main peaks are indicated. The Si signal originates from the substrate and is particularly visible for thin samples. (**B**) XPS spectra of different quantum dots in the Cd3d and N1s regions.

**Figure 6 nanomaterials-14-00010-f006:**
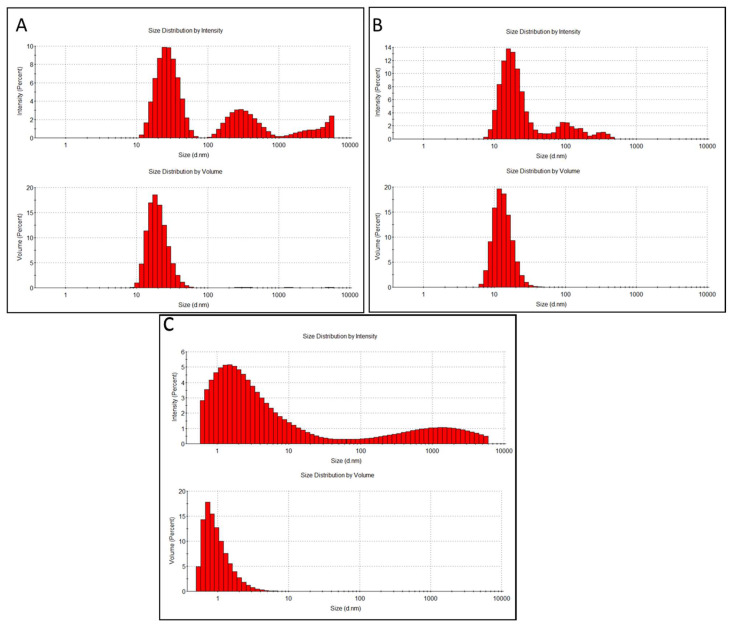
Size distributions of CdSe/ZnS QDs (**A**), CdSe/ZnS-Cys QDs (**B**), and CQDs (**C**), based on the dynamic light scattering intensity (upper graph in each panel) and particle volume (bottom graph in each panel). The hydrodynamic sizes were measured in MilliQ water using ZetaSizer Nano ZS (Malvern). The average hydrodynamic diameters for each type of QDs (Table 1) were calculated by the instrument software based on the dynamic light scattering intensity distributions.

**Figure 7 nanomaterials-14-00010-f007:**
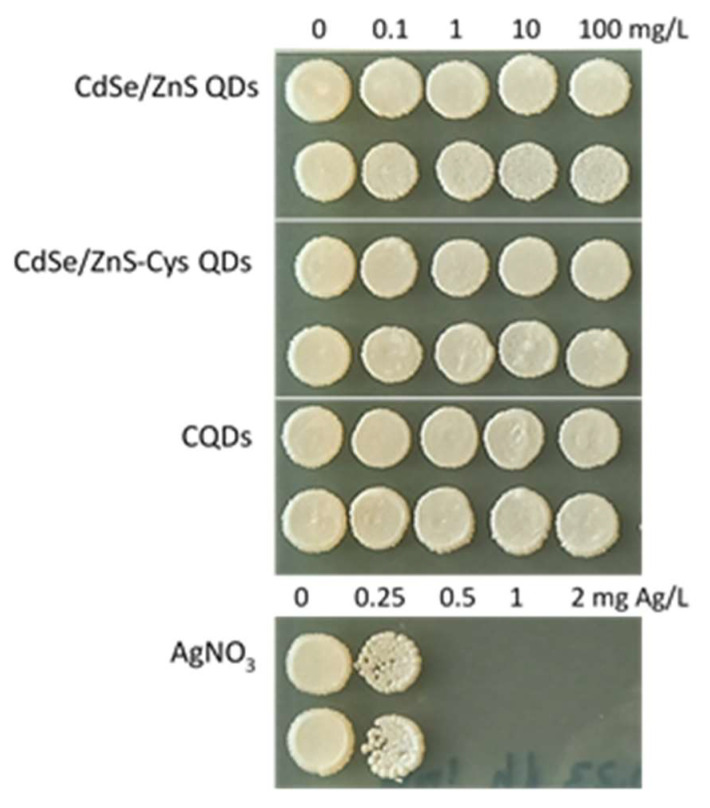
A viability assay (spot test) showing the colony-forming ability of yeast *S. cerevisiae* BY4741 after 24 h of exposure to CdSe/ZnS quantum dots (QDs), CdSe/ZnS QDs connected to cysteine (Cys), and carbon quantum dots (CQDs) and AgNO_3_ (as a positive control) in MilliQ water at 30 °C. Two replicates per tested compound were presented.

**Figure 8 nanomaterials-14-00010-f008:**
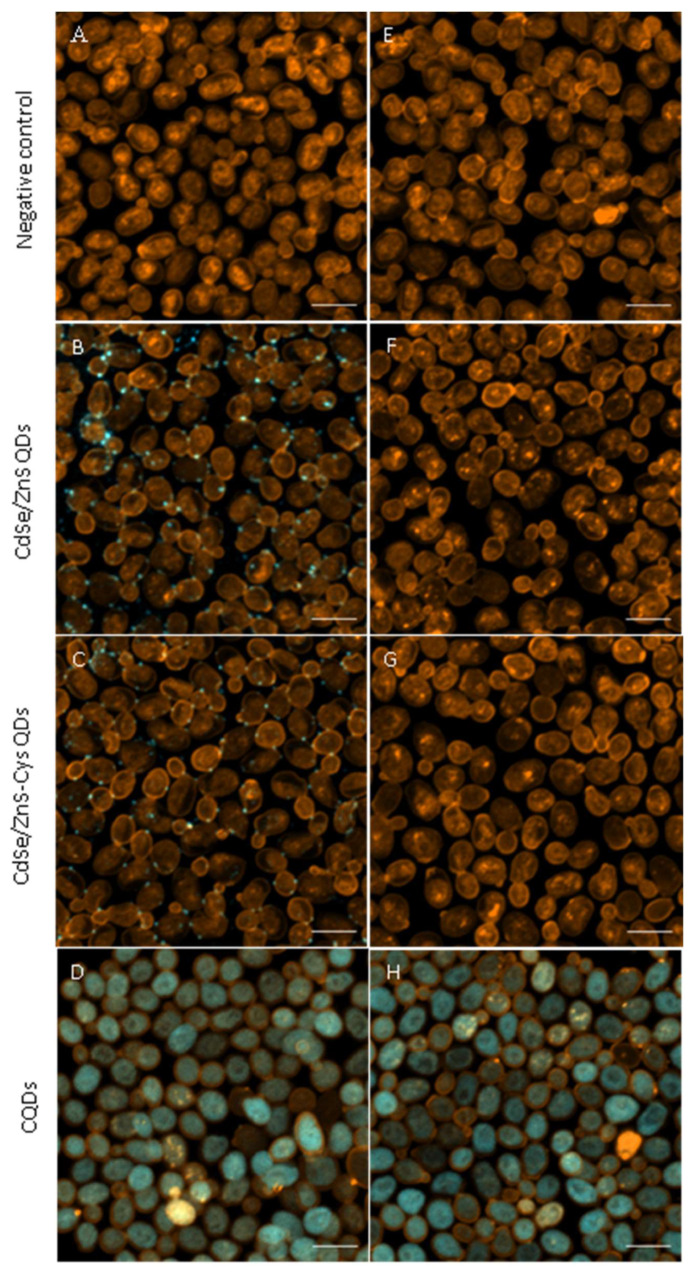
Confocal laser scanning microscopy (CLSM) images of the cellular interactions of quantum dots (QDs) in *S. cerevisiae* BY4741 cells stained with CellBrite Fix 555 membrane stain (orange pseudocolour) were incubated with QDs (blue pseudocolour) for 24 h. After the incubation, cells were either not washed (**A**–**D**) or washed (**E**–**H**) with MilliQ water to assess the strength of interactions. Scale bars correspond to 5 µm.

**Table 1 nanomaterials-14-00010-t001:** Primary and average hydrodynamic diameters (Z-average diameters, nm), intensity-weighted particle size distribution (main peaks and their intensity%), polydispersity indices (PdI), and zeta potentials (mV) of CdSe/ZnS quantum dots (QDs), the same QDs conjugated to cysteine and carbon quantum dots (CQDs) in MilliQ water. Values are the average of three measurements ± the standard deviation.

Sample	Pristine Diameter, nm	Hydrodynamic Diameter, nm **	Average Sizes of Main Peaks, nm (% Intensity) ***	PdI	Zeta Potential, mV
CdSe/ZnS QDs 100 mg/L	10 *	47.4 ± 2.4	Peak1 28.3 ± 9.7 (66%) Peak2 347 ± 140 (31%) Peak3 5245 ± 447 (3.8%)	0.50 ± 0.05	−70.0 ± 2.4
CdSe/ZnS—Cys QDs 100 mg/L	15–20 *	308 ± 150	Peak1 17.4 ± 5.9 (77%) Peak2 67.7 ± 6.5 (6.4%) Peak3 368 ± 94 (9.1%)	0.43 ± 0.05	−64.7 ± 4.0
CQDs 500 mg/L	2–3	6.08 ± 3.35	Peak1 1.98 ± 1.16 (59%) Peak2 1425 ± 1359 (41%)	0.43 ± 0.16	−19.4 ± 12.0

* reported by the manufacturer, ** Z-average diameters calculated by ZetaSizer Nano ZS software based on the particle size distribution by intensity, *** intensity-weighted particle size distribution (upper graphs in panels A, B, and C).

## Data Availability

Data is contained within this article and its Appendix A.

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
