# Peer review of "Comparison of Toxicity and Cellular Uptake of CdSe/ZnS and Carbon Quantum Dots for Molecular Tracking Using Saccharomyces cerevisiae as a Fungal Model"

_nanomaterials, 2023, doi:10.3390/nano14010010_

Round 1
Reviewer 1 Report
Comments and Suggestions for Authors
The manuscript details the comparative uptake and toxicity of semiconductor quantum dots of either CdSe or amorphous carbon. Though I appreciate the value of the studies aim of looking at CQDs as probes for fungal systems, I do not believe the results support the conclusions/interpretations that the authors propose.
The principal issue is that the CdSe QDs are clearly aggregating upon modification and changing size/charge, etc. Therefore, the conclusion that the core is somehow modifying the uptake seems false, when it is more likely that size is the main culprit. Furthermore, I have issue with the presentation of an average size taken from DLS data. It is known that intensity response from DLS data is very skewed towards larger sized particles, I would look at representing the various populations or using something like Number or Volume to get a better representation of the particles sizes.
I also believe the work of Bilal et al is relevant (https://onlinelibrary.wiley.com/doi/full/10.1002/smll.201900510) to your manuscript and would consider citing it.
Reviewer 2 Report
Comments and Suggestions for Authors
In this manuscript, “Comparison of Toxicity and Cellular Uptake of CdSe/ZnS and Carbon Quantum Dots for Molecular Tracking Using Saccharomyces cerevisiae as a Fungal Model” by Färkkilä et al. reports the synthesis of amino acid-based carbon quantum dots and compared their toxicity and uptake with commercial CdSe/ZnS QDs that conjugated with the amino acid cysteine (Cys) using yeast Saccharomyces cerevisiae as a proxy for mycorrhizal fungi. Although, some preliminary results are demonstrated, some results seem quite strange. Therefore, I would suggest that authors may take at least a revision. Here are the comments and suggestions:
1. Please add the results of this work to Table S1 for easy comparison.
2. In Fig. 1, what would the black dash line stand for?
3. In Fig. 2, please add results of other samples.
4. Abbreviations should be defined before their first use.
5. In Fig. 5, peaks for Cd or Zn in CdSe/ZnS-cys seem disappeared.
6. The conclusions can be extended.
Round 2
Reviewer 1 Report
Comments and Suggestions for Authors
Since it is basically the same paper my opinion holds. Since the editor chose to allow the paper to merely have minor modifications and be re-submitted I assume he did not truly consider my opinion. As such I wish both the authors and editors best of luck and withdraw from engaging with this work further.
Reviewer 2 Report
Comments and Suggestions for Authors
1. The absorbance spectra of CQD seems quite strange, please see DOI:10.1039/C5GC00686D.
2. Please add results of other QDs to Figs. 1 and 2.
3. X-ray diffraction of these samples should be added.
4. Authors should reduce the thickness of L-Cys on CdSe/ZnS-Cys to get the similar size for the comparison the toxicity of other samples.
5. Results of this work in Table S1 should be separated.
